# Tuberculosis care quality in urban Nigeria: A cross-sectional study of adherence to screening and treatment initiation guidelines in multi-cadre networks of private health service providers

Lauren A. Rosapep[1]*, Sophie Faye[1], Benjamin Johns[1], Bolanle Olusola-Faleye[2], Elaine M. Baruwa[1], Micah K. Sorum[1], Flora Nwagagbo[2¤], Abdu A. Adamu[3], Ada Kwan[4,5], Christopher Obanubi[4¤], Akinyemi Olumuyiwa Atobatele[6]

1 Abt Associates Inc., International Development Division, Rockville, MD, United States of America, 2 Abt Associates Inc., Sustaining Health Outcomes through the Private Sector (SHOPS) Plus Project, Lagos, Nigeria, 3 Abt Associates Inc., Sustaining Health Outcomes through the Private Sector (SHOPS) Plus Project, Kano, Nigeria, 4 Division of Pulmonary and Critical Care, University of California San Francisco, San Francisco, CA, United States of America, 5 Division of Health Policy and Management, School of Public Health, University of California, Berkeley, CA, United States of America, 6 U.S. Agency for International Development (USAID), Office of HIV/AIDS and Tuberculosis, Abuja, Nigeria

¤ Current address: U.S. Centers for Disease Control and Prevention (CDC), Abuja, Nigeria
* lrosapep@gmail.com

## Abstract

Nigeria has a high burden of tuberculosis (TB) and low case detection rates. Nigeria's large private health sector footprint represents an untapped resource for combating the disease. To examine the quality of private sector contributions to TB, the USAID-funded Sustaining Health Outcomes through the Private Sector (SHOPS) Plus program evaluated adherence to national standards for management of presumptive and confirmed TB among the clinical facilities, laboratories, pharmacies, and drug shops it trained to deliver TB services. The study used a standardized patient (SP) survey methodology to measure case management protocol adherence among 837 private and 206 public providers in urban Lagos and Kano. It examined two different scenarios: a "textbook" case of presumptive TB and a treatment initiation case where SPs presented as referred patients with confirmed TB diagnoses. Private sector results were benchmarked against public sector results. A bottleneck analysis examined protocol adherence departures at key points along the case management sequence that providers were trained to follow. Except for laboratories, few providers met the criteria for fully correct management of presumptive TB, though more than 70% of providers correctly engaged in TB screening. In the treatment initiation case 18% of clinical providers demonstrated fully correct case management. Private and public providers' adherence was not significantly different. Bottleneck analysis revealed that the most common deviations from correct management were failure to initiate sputum collection for presumptive patients and failure to conduct sufficiently thorough treatment initiation counseling for confirmed patients. This study found the quality of private providers' TB case

**Data Availability Statement:** All data relevant to the study are freely available on USAID's Development Data Library (DDL) at: https://data.usaid.gov/Tuberculosis/Nigeria-Tuberculosis-Program-Quality-of-Care-Study/f8bn-qej9 The DDL is a public repository maintained by USAID. Anyone can access this data at no cost once they have registered for a free user account on the DDL.

**Funding:** Authors LAR, SF, BJ, BOF, EMB, MKS, FN, AAA, and AK implemented this work with funding provided by the US Agency for International Development (USAID) under the SHOPS Plus project (Contract #: AID-OAA-A-15-00067, https://www.usaid.gov/). All of these authors were affiliated with the SHOPS Plus project and not direct recipients of a USAID contract. At the time of the research authors CO and AOA were employees of the funding agency USAID, and provided inputs into the study design, endorsed the decision to publish results, and provided critical review of the draft manuscript and approved the submitted manuscript.

**Competing interests:** I have read the journal's policy and the authors of this manuscript have the following competing interests: authors AOA and CO were employees of USAID, the funder of the research and of the SHOPS Plus project; authors BOF, AAA, and FN were employees of the SHOPS Plus project in Nigeria, which implemented the intervention studied in this research; the SHOPS Plus project is implemented by Abt Associates globally and authors LAR, SF, BJ, EMB, and MKS are employees of Abt Associates.

management to be comparable to public providers in Nigeria, as well as to providers in other high burden countries. Findings support continued efforts to include private providers in Nigeria's national TB program. Though most providers fell short of desired quality, the bottleneck analysis points to specific issues that TB stakeholders can feasibly address with system- and provider-level interventions.

## Introduction

With 440,000 cases of TB in 2019, Nigeria is a high burden country for tuberculosis (TB), multidrug resistant TB (MDR-TB), and TB-HIV [1]. The country has one of the lowest case detection rates among high TB burden countries with 117,320 (27 percent) of the incident cases being notified in 2019 [1], which both inhibits infection control measures and leads to treatment initiation delays. In addition, a recent national assessment of TB services for confirmed TB patients found that health facilities lack consistent access to essential equipment and drug supplies and health workers do not receive refresher trainings and frequent supervision [2]. Together these data suggest that the existing health system in Nigeria faces multiple challenges in the management of presumptive and confirmed TB.

Historically, Nigeria's national program to screen and treat TB was limited to the public sector [3]. Since the majority of Nigerians report that they seek health care from the private sector [4–6], and the WHO's STOP TB strategy recommends linking private facilities to the national TB program, Nigeria's National TB and Leprosy Control Programme (NTBLCP) and its donor partners have recently sought to expand private sector participation in Nigeria's TB response [7,8].

Outside of Nigeria, most studies examining private sector TB service delivery in LMICs including Ethiopia, India, Kenya, the Philippines, and Thailand, have identified important shortcomings in effectiveness of tuberculosis cases management by private clinicians and pharmacists [9–17]. Within Nigeria, several studies have examined treatment success rates (TSR) in clinical facilities participating in small-scale PPM pilots in Anambra, Imo, Kaduna, Lagos, Ogun, and Plateau states [7,18–23]. Although these studies found that observed TSR for private sector patients was lower than Nigeria's national treatment success target of 90 percent, these rates were often comparable to TSR reported for public sector patients in those states [7,18–23], which supports continued scale up of PPM in Nigeria to increase access to care.

One of the largest efforts to scale up PPM in Nigeria to date is the United States Agency for International Development (USAID)-funded Sustaining Health Outcomes through the Private Sector (SHOPS) Plus program, which began implementing a public-private mix (PPM) approach in 2018 to intensify private sector TB case detection and treatment in the urban areas of Lagos and Kano states. SHOPS Plus's PPM approach builds on models used in India [24], and is the first effort in Nigeria to develop, train and equip large networks of private, multi-cadre facilities to screen, diagnose, and treat TB.

The SHOPS Plus network model's ability to provide quality services is currently untested, and it is not well understood whether enhancing multiple cadres of private providers' TB knowledge and management capacity contributes to overall improvements in TB quality of care in Nigeria. Existing studies of PPM care quality [7,18–23] in Nigeria have focused primarily on treatment outcomes, rather than screening and treatment initiation, and have also exclusively focused on clinical providers. To fill this gap, we used a standardized patient (SP) (also known as the "simulated patient" or "mystery client") survey method to rigorously examine

the levels of quality of presumptive and newly diagnosed TB patient management in the private sector of two Nigerian states, the variation across facility cadres, and to assess differences between providers in a private sector program compared to public sector providers.

Although SP surveys have previously been used to assess the quality of family planning and reproductive health services [25–28], this is the first study to use SPs to examine TB care quality in Nigeria. Since the SP method in this study used adapted instruments and approaches from other recent SP studies examining TB care quality in India [12–14,16,17], Kenya [15], China [29], and South Africa [30,31] this study also contributes to the literature on quality of TB care in high-burden settings by producing comparable data that can be similarly interpreted.

Further, where these studies examine a lenient, binary measure of correct case management for TB based primarily on a single key outcome (e.g., initiating a TB diagnostic for a patient with presumptive TB or prescribing TB treatment to a confirmed TB patient), this is the first study among the quality of TB care literature that conceptualizes correct case management not just as the achievement of a single outcome or set of independent outcomes, but as a combination of actions that are required if providers are to adhere to the protocols and diagnostic algorithms defined in NTBLCP's guidelines for TB care [32]. By quantitatively capturing the "bottlenecks", or points in the case management flow where a disproportionate number of providers depart from protocol adherence, we were better able to examine case management not just in terms of correct actions, but account for decisions that result in unnecessary, inefficient, or even potentially harmful actions. These actions can have individual and community-level impacts, for example ordering (and incurring the cost of) a TB diagnostic in the absence of appropriate screening or prescribing medications that can mask TB symptoms or fuel antimicrobial resistance. By using Sankey diagrams to visually present results of this analysis we can more easily depict TB providers' case management strengths and weaknesses, which in turn can help programs supporting the delivery of quality TB services better understand what to emphasize and monitor when targeting, designing, or improving training and technical assistance mechanisms for providers.

## Background and study setting

We purposively selected urban areas of Lagos and Kano states for this study, as these were the only areas in which SHOPS plus was implementing its network model in 2019 when study activities commenced. Lagos is Nigeria's most economically prosperous metropolitan area, with an estimated population of over 20 million and approximate per capita income of $5,000 per year [33]. Kano is the largest metropolitan area in Northern Nigeria, with an estimated population of 3.9 million and approximate per capita income of $1,200 per year [34].

In both states SHOPS Plus networks are comprised of four different health service cadre: clinical facilities (including clinics, nursing homes, and hospitals), stand-alone private laboratories, private community pharmacies, and retail shops that are formally licensed by the state ministry of health to sell over-the-counter medications (known locally as patent and proprietary medicine vendors, or drug shops). These providers are trained in appropriate TB screening, diagnosis, and treatment practices, and then organized into "hub-and-spoke" clusters. Member laboratories, pharmacies, and drug shops serve as the "spokes" and drive patient traffic into hubs by screening clients for TB, collecting sputum samples from presumptive patients for GeneXpert testing in government and private laboratories, and/or referring presumptive patients to a clinical facility for further assessment or treatment. Clinical facilities serve as hubs, which receive referrals from spokes, screen their own client base for TB, and treat confirmed, drug-susceptible TB (DS-TB) patients using NTBLCP-supplied drugs.

Prior to forming these networks SHOPS Plus conducted rapid assessments of 1,054 private sector facilities in Kano and 1,622 in Lagos in 2018 to assess current availability of TB services, suitability to provide these services, and facility willingness to participate in TB service provision and capacity to offer TB services. Facilities were identified using the Nigeria Health Facility Registry (NHFR) [35], additional lists of private clinical providers provided by state governments; member rosters provided by professional laboratory, pharmacy, and drug shop associations. Since none of these lists were believed to provide a comprehensive picture of the private facility landscape, snowball sampling was required to identify additional facilities, particularly non-clinical facility types.

Willing facilities were then invited to send up to two staff members to a cadre-specific training that was jointly hosted by SHOPS Plus and the NTBLCP. All trainings used a modified version of NTBLCP's national training curriculum and were aligned to NTBLCP guidelines. Staff from private clinics, health centers, nursing homes and hospitals attended a five-day clinical facility training focused on screening, diagnosis, care, and treatment, including complications management, TB/HIV integration, the importance of linkages between directly observed treatment short-course (DOTS) and drug-resistant TB (DR-TB), the use of chest x-rays for TB screening. Laboratory training was also 5-days and focused on screening and referrals in addition to instruction on acid-fast bacilli (AFB) microscopy TB detection. The one-day training for pharmacies and drug shops focused on screening, how to take sputum samples, and referrals.

After training SHOPS Plus grouped facilities into networks based on physical proximity within a specific local government area (LGA). SHOPS Plus initially set up networks in 19 of Lagos State's 20 LGAs and 11 of the LGAs that include and surround the Kano city metropolitan area. Networks were intended to be flexible in nature and grow over time as the project conducted additional outreach to private providers. By early 2019, SHOPS Plus had engaged 1,598 private facilities in the two states (1,055 in Lagos and 543 in Kano). Based on the total number of private clinical facilities listed in the Nigeria Health Facility Register as of January 2019 (1,808 in Lagos and 261 in Kano), it appears that approximately 23% (n = 408) Lagos clinical providers and 36% (n = 94) of Kano city metropolitan area clinical providers were part of a SHOPS Plus network. Prior to SHOPS Plus there were about 100 other private clinical facilities participating in the Lagos state TB program, and no private facilities participating in the Kano program.

Following training and inclusion in a network SHOPS Plus provided continual support and oversight to facilities to ensure the availability of high-quality services and commodities through regular (at least bi-weekly) supportive supervision visits. Additionally, laboratories, pharmacies, and drug shops receive an incentive payment of N200 (USD~$0.52) for each presumptive patient referred for clinical or laboratory diagnosis and an additional N1000 (USD~$2.61) for each presumptive patient that is diagnosed as positive for TB. Using NTBLCP-supplied reagents, laboratories can conduct AFB sputum testing (and charge up to N500 (USD ~ $1.31)) in addition to collecting sputum samples for free testing in public or private laboratories with GeneXpert machines. SHOPS Plus clinical facilities receive no incentive payments and cannot charge for TB drugs, which are provided free of charge by the government. They may optionally charge consultation and service fees up to a total maximum charge of N35,000 (~$91 USD) in Lagos and N20,000 (~$52 USD) in Kano over six months of treatment, however many network facilities choose not to charge any treatment-related fees.

## Methods

This study used the SP survey method, which deploys trained data collectors to pose as incognito patients with a standardized case presentation to health facilities, after which they debrief

the encounter with a field supervisor using structured exit questionnaire administered to generate the quality-of-service-provision data. The SP method is considered a rigorous approach to measure the quality of initial patient-provider interactions across various health areas and disciplines [13,36].

## Facility sampling

As of March 2019, SHOPS Plus had engaged 1,598 private facilities in the two states; In addition to the 1,598 private facilities engaged in SHOPS Plus networks (1,055 in Lagos and 543 in Kano) there were a total of 304 state-run public facilities in Lagos (including 195 centers offering TB treatment using the DOTS protocol and 290 state-run public facilities (including 190 DOTS centers) in the 11 Kano LGAs where SHOPS Plus operates networks. To construct sampling frames, we obtained network facility lists from SHOPS Plus and downloaded public facility lists from the Nigeria Health Facility Registry [35]. Facilities were treated as the sampling unit. The private facility sampling frame was stratified by state (Kano and Lagos) and then by each of the four provider types (clinical provider, stand-alone laboratory, pharmacy, and drug shop) resulting in eight substrata. The public facility sampling frame was stratified by state (Kano and Lagos) and then by DOTS status (clinical facilities participating in the state TB program (DOTS sites) and clinical facilities not yet engaged by the state TB program (non-DOTS sites)) resulting in four substrata. Following sample size calculations for each substratum, we randomly sampled a total of 903 private facilities and 207 public facilities for inclusion in the study. We used a 95% level of confidence ($\alpha = 0.05$) for all calculations, and the set the proportion of providers following appropriate patient management protocols at p = 0.5 to maximize variance. Resource limitations and programmatic priorities were considered when determining the margin of error (MOE) for each substratum. For private facilities in the SHOPS Plus network, the MOE is ±5%, for public facilities the MOE is ±10%. Further details on the study sample are provided in S1 Annex.

## Assessment of quality using two SP case scenarios

We designed two case scenarios to test whether health care personnel in sampled facilities were appropriately managing walk-in presumptive and confirmed TB patients. These scenarios were informed by previous SP studies [13–17,29–31] and then customized with input from TB policy and clinical experts affiliated with USAID/Nigeria, the Lagos and Kano state TB programs, and the SHOPS Plus technical team to ensure the cases would test provider alignment with NTBLCP guidelines [32]. Other members of the public and TB patients were not engaged in study design.

In the first scenario (Case 1) SPs make an unannounced visit to a facility and present with presumptive TB symptoms—a chronic cough and fever that was not resolving, similar to the "textbook" case used in other recent SP studies [13,15,17,29,31]. In laboratories SPs presented with only a chronic cough (rather than a fever) and directly requested a test "to see what was wrong with them". These modifications were introduced to ensure SPs presented plausibly (since clients in private laboratory settings do not typically expect to be examined or treated for an illness) and were not subject to invasive blood draws for malaria or typhoid diagnostics, since fever is often associated with these those two illnesses. Overall, SPs' Case 1 presentation should prompt trained providers to screen for TB and initiate further diagnostic testing (i.e., GeneXpert, AFB microscopy, and/or x-ray).

In the second scenario (Case 2) the SP has been previously "identified" as presumptive for TB in a private pharmacy, given a positive GeneXpert test result, and referred to clinical facility for follow up. Case 2 is designed to explore whether providers interacting with confirmed

**Table 1. Standardized patient scenarios.**

| Case | Type of facilities implemented | Presenting condition (SP opening statement) | Expected management |
|---|---|---|---|
| Case 1: presumptive patient scenario | Private SHOPS Plus facilities (clinical, stand-alone labs, pharmacies, drug shops) Public facilities (DOTS-designated, non-DOTS-designated) | Clinical facilities, pharmacies, drug shops: *"I am having fever and cough that is not getting better."* Labs: *"I am having cough that is not getting better. What test can I do to figure out what is wrong with me?"* | Provider is prompted to initiate screening, initiate further diagnostic testing (by requesting a sputum sample, recommending an x-ray, or referring patient elsewhere for diagnostics if the facility is not participating in sputum sampling). In absence of a diagnostic result, the provider refrains from dispensing inappropriate (i.e., anti-TB drugs, fluoroquinolones, steroids) or unnecessary (i.e., other antibiotics) medications. |
| Case 2: treatment initiation scenario | Clinical facilities (SHOPS Plus clinical and public DOTS facilities) | *"I visited the pharmacy and they did a test and told me to come here [show GeneXpert test report]."* | Upon examining the GeneXpert report to confirm the diagnosis for DS-TB, the provider is expected to provide counseling for new TB patients and ask the patient to identify a treatment supporter. May immediately move to initiate treatment or ask the patient to return to facility with treatment supporter to initiate treatment. Should not prescribe unnecessary or inappropriate medications (i.e., fluoroquinolones, steroids, or non-TB antibiotics). |

patients tested and identified within the SHOPS Plus network would engage in TB counseling and initiate treatment according to NTBLCP guidelines.

Table 1 includes additional detail on the cases used in this study.

## Ethical approval

Approvals were obtained from the Kano State Ministry of Health (MOH/Off/797/T.1/1179) and the Lagos State University Health Research and Ethics Committee (LREC/06/10/1166). All individuals who participated as SPs were trained to protect themselves from any invasive tests or procedures (such as blood draws or injections). For this study, the study team requested and was granted an unconditional waiver of facility and provider informed consent to uphold the validity of the study data.

## Data collection

To ensure SPs presented and responded to providers in a believable and undetectable manner we used publicly available and validated manuals and tools to rigorously prepare SPs for field deployment [37,38]. After a five-day pilot, the SPs made unannounced visits to all sampled facilities to implement Case 1. SHOPS Plus clinical facilities and public DOTS facilities received one additional unannounced visit from SPs implementing Case 2. To prevent detection, different SPs were assigned to present Case 1 and Case 2. Directly after each interaction, a field supervisor debriefed the SPs using a structured exit questionnaire to record details of their interaction. All SP visits were completed between 10 June and 12 July 2019.

## Outcomes definition and analysis

Our main outcome of interest is the binary outcome correct management, which is motivated by the NTBLCP's expectation that any presumptive or confirmed TB patient that walks into a trained facility should be appropriately managed regardless of the individual health personnel with whom they interact. For Case 1, "correct management" included three elements, which are informed by the sequence of actions indicated by the diagnostic algorithms defined in the NTBLCP's guidelines: 1) confirmation of core TB symptoms (through patient history taking); 2) engaging in sputum collection to enable appropriate diagnostic testing (or referring the SP

elsewhere for diagnostics); and 3) refraining from prescribing medicines deemed inappropriate (including TB drugs, fluoroquinolones, and steroids) or unnecessary (other broad or narrow-spectrum antibiotics not including TB drugs or fluoroquinolones) for undiagnosed presumptive patients.

For Case 2, "correct management" included five elements, based on the sequence of actions that the NTBLCP guidelines specify that providers should take when first engaging with a newly-confirmed TB patient: 1) using the GeneXpert report to confirm the patient's TB diagnosis; 2) providing sufficient TB counseling (explaining TB disease and its treatment); 3) requesting the patient identify a treatment supporter; 4) refraining from prescribing medicines deemed inappropriate or unnecessary to dispense to a confirmed TB patient (including fluoroquinolones, steroids, or other broad- or narrow-spectrum antibiotics); and 5) initiating TB treatment.

To be counted as correct, Case 1 and Case 2 interactions required demonstration of all the elements defined for the case. In both Case 1 and Case 2 we chose to count instances where providers prescribe fluroquinolones and steroids (which can mask TB symptoms and hinder TB diagnosis or treatment), as well as the prescription of other broad- or narrow-spectrum non-TB antibiotics as incorrect management, since prescription of antibiotics in the absence of a lab-confirmed diagnosis is generally recognized by public health practitioners as a factor contributing to community-level antibiotic resistance. Further details on the criteria for correct management and data collection are provided in S1 Annex file.

For each facility cadre, we calculated means or proportions along with 95% confidence intervals (CI) for all the component elements described above, and the composite measure of correct management defined for each case. For results calculated across one type of facility, no sample weights were employed because simple random sampling was used to select facilities within the substrata. When calculating results for multiple facility types, we estimated sample weights based on average patient screening volumes for June and July 2019 for each facility type. With this weighting, the results reflect average care provided to a patient with that case's presentation as opposed to the care provided by the average provider. All results used linearized standard errors appropriate for a survey design.

Since measurement of the correct management outcome is binary and reflects only the proportion of providers who meet all defined case elements, and individual case element measures are agnostic to other provider actions taken during the SP interaction, looking at results for these measures alone does not give a full picture of how providers were completing case management elements in combination and in the implied sequence recommended in the national guidelines (e.g., the proportion of providers who screen and order a diagnostic test, or the proportion of providers who confirm TB and also provide basic counseling on TB and its treatment). To analyze the provider decision making process on the case management outcome, we use a bottleneck analysis to show the flow of actions that comprise the main outcome of interest, as well as others that play a role in ideal TB case management for the specific cadre. For each case we referenced the NTBLCP guidelines to determine the sequencing of elements and defined a "bottleneck" as an element in the case management sequence where a proportion of providers who fail to align with guidelines is disproportionate (e.g., greater than 10 percentage points) to the proportion who do not align with elements in other points of the sequence. For example, along a case management sequence where providers must meet three elements to be considered to have correctly managed a patient, if the proportion of providers who fail to demonstrate the second (or third) element in the sequence is much larger than the proportion failing to meet the preceding element, this may signal that there is a particular part of the case management sequence that is posing a disproportionate barrier to achievement of correct management, and thus may require extra remediation or monitoring to improve the rates of overall case management success.

To assess differences in case management between different SHOPS Plus cadres in the presumptive patient scenario, we used logistic regressions to assess facility, provider, and SP characteristics on the probability of correct management for both scenarios. Covariates were limited to what we could observe about the facility and the provider during the interaction with the SP and included: a constructed interaction variable reflecting the facility type and the most senior-level provider that attended to the SP, facility location (state), a constructed interaction variable reflecting the gender of the SP and the gender of the most senior provider that attended the SP, and a categorical variable reflecting the average number of real patients waiting in the facility before and after the SP interaction (as an index of patient caseload). Regression results are reported as adjusted odds ratios (ORs) for each variable. Finally, to contextualize the results for the private SHOPS Plus network facilities, we conducted a secondary analysis using significance testing to benchmark SHOPS Plus results against a proxy for the existing quality of TB care in Nigeria. We used results from the public facilities as our proxy for the existing standard of care, given that prior to SHOPS Plus, most TB services in Nigeria were provided in these settings. We used proportion tests to compare correct management (yes or no) between types of providers at alpha = 0.05 level of significance.

All statistical analyses were performed using Stata 16 software.

## Results

We report data on a total of 1,390 completed interactions across 1,043 representatively sampled public and private facilities. Table 2 shows the total number of facilities that were part of the sampling frame for each case, the number of SP visits expected, and the number of SP visits completed across all facility cadres and sectors.

SPs completed 93% (n = 837) of expected Case 1 attempts in private facilities and 99% (n = 206) in public facilities. For Case 2, SPs completed 81% (n = 228) of private facility attempts and 84% (n = 119) of public facility attempts. There were several reasons SPs were unable to complete interactions (n = 85 in Case 1, n = 77 in Case 2) including: SP inability to

**Table 2. Study sample and completed SP visits.**

| Scenario | Facility Sector | Cadre | LAGOS FACILITIES | | | KANO FACILITIES | | | ALL FACILITIES | | |
|---|---|---|---|---|---|---|---|---|---|---|---|
| | | | Total | Expected visits | Completed Visits | Total | Expected visits | Completed Visits | Total | Expected visits | Completed Visits |
| Case 1: Presumptive Patient Scenario | Private Facilities in SHOPS Plus network | Clinical Facilities | 428 | 203 | 196 | 100 | 80 | 73 | 528 | 283 | 269 |
| | | Labs | 104 | 83 | 80 | 30 | 28 | 20 | 134 | 111 | 100 |
| | | Pharmacies | 136 | 101 | 94 | 19 | 19 | 19 | 155 | 120 | 113 |
| | | Drug shops | 387 | 194 | 171 | 394 | 195 | 184 | 781 | 389 | 355 |
| | | Sub-Total | 1055 | 581 | 541 | 543 | 322 | 296 | 1598 | 903 | 837 |
| | Public Facilities | DOTS Facilities | 195 | 71 | 72 | 190 | 70 | 70 | 385 | 141 | 142 |
| | | "Non-DOTS" Facilities | 109 | 27 | 23 | 100 | 39 | 41 | 209 | 66 | 64 |
| | | Sub-Total | 304 | 98 | 95 | 290 | 109 | 111 | 594 | 207 | 206 |
| Case 2: Treatment Initiation Scenario | Private Facilities in SHOPS Plus network | Clinical Facilities | 428 | 203 | 174 | 100 | 80 | 54 | 528 | 283 | 228 |
| | Public Facilities | DOTS Facilities | 195 | 71 | 60 | 100 | 70 | 59 | 295 | 141 | 119 |
| | | Sub-Total | 623 | 274 | 234 | 200 | 150 | 113 | 823 | 424 | 347 |
| | | **Total** | **1359** | **953** | **870** | **833** | **581** | **520** | **2192** | **1534** | **1390** |

find or initiate an interaction in a facility due to prolonged facility closure or provider absence after multiple visit attempts (n = 80 in Case 1, n = 27 in Case 2), the facility requested to re-test the SP for TB before proceeding with treatment initiation (n = 37 in Case 2 only), or the facility was no longer engaging with TB patients (n = 1 in Case 1, n = 4 in Case 2). In a small number of attempts SPs unexpectedly interacted with an individual identifying as a SHOPS Plus staff member (n = 1 in the Case 1, n = 5 in Case 2) or the SP noted that the provider had regarded them with suspicion or accused them of faking their illness to obtain drugs for the black market (n = 1 in Case 1, and n = 4 in Case 2). Incomplete interactions were not included in analysis. S2 Annex provides further details on response rates.

The following sections present key case results rounded to the nearest whole number followed by a 95% confidence interval (CI) in brackets. Tables A-C in S3 Annex provide detailed, state-specific results for Case 1; Table A in S4 Annex provides detailed, state-specific results for Case 2.

## Case 1 results: Management of presumptive patients with "textbook symptoms"

Except for stand-alone laboratories (83% [80%-86%]), a minority of private clinical facilities (31% [28%-35%]), drug shops (21% [19%-25%]), and pharmacies (8% [6%-10%]) across both states met all three elements needed to correctly manage Case 1. Although the proportion of providers meeting all three elements across most cadres is low, providers were more successful in meeting each of the individual case elements (Fig 1). This is especially true for the first element (screening) as a large majority of providers (70% or more) in each cadre questioned the SP about the nature of their cough. For the other two elements—related to sputum sampling and refraining from inappropriate or unnecessary medicine dispensing—there are more modest rates of success across cadres.

In general, dispensing of one or more medicines was common outside of laboratory settings, especially among pharmacies (73% [69%-76%]) and drug shops (69% [66%-73%]). Medicine dispensing was less common in clinical facilities (42% [39%-46%]). Among those dispensing or prescribing any medicine, the most-prescribed medicines include non-TB

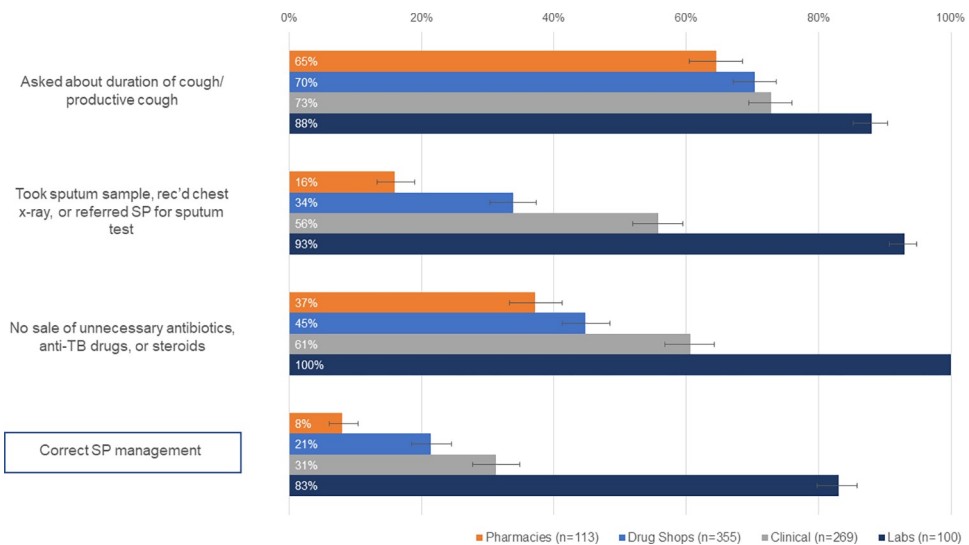

**Fig 1. Case 1: Management of SPs presenting as presumptive patients, by facility type.**

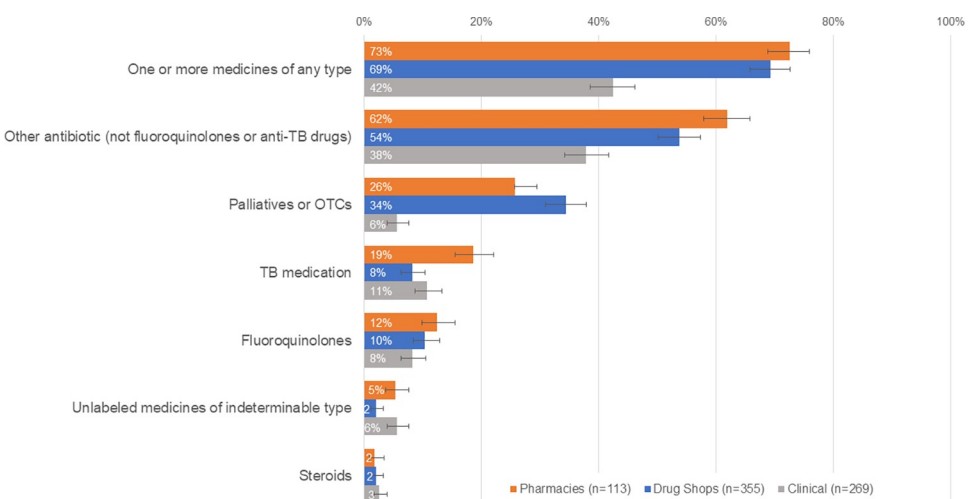

**Fig 2. Case 1: Medicines dispensed to SPs presenting as presumptive patients, by facility type.**

antibiotics and palliative or over-the-counter (OTC) treatments. Few private providers dispensed medicines that are considered inappropriate to dispense to a presumptive patient prior to receipt of a TB diagnostic result, including TB medications, fluoroquinolones, or steroids (Fig 2). Recognizing that TB is still a relatively rare illness in Nigeria and that providers are motivated to provide services that they feel are in the best curative interest of their patients, we acknowledge that some providers may feel clinically justified to prescribe other non-TB antibiotics to their patients while they wait for the results of their TB diagnostics. To account for this reality, we conducted a sensitivity analysis to explore the impact of relaxing the criterion related to inappropriate and unnecessary prescribing on overall correct management outcomes. We found that Case 1 results are robust to this sensitivity analysis, which indicates other provider practices not related to medicine prescribing are driving incorrect management of presumptive patients. See Tables D-G in S3 Annex for sensitivity analysis results.

The bottleneck analysis sequence for Case 1 (Fig 3) starts with starts with screening for TB and continues with the initiation of (or referral for) an appropriate diagnostic test, as it would be inappropriate to order a TB diagnostic without also confirming that a patient was experiencing TB symptoms. Although relatively high percentages of providers of all cadres engaged in TB screening, a bottleneck emerges at the next step in the sequence for non-clinical providers. The bottleneck depicted at the second stage of the case management flow in Fig 3 shows that even though they successfully screened, large proportions of drug shops (60%), and pharmacies (80%) failed to initiate diagnostic testing, that is, did not obtain a sputum sample or refer the SP elsewhere for a TB diagnostic. Among the providers who successfully screened and initiated a TB diagnostic for the patient a majority (laboratories 100%; drug shops 75%; clinical facilities 65%; pharmacies 60%) also refrained from dispensing unnecessary or inappropriate medicines and ultimately achieved correct management. To contextualize the results of the bottleneck analysis we conducted a correlation analysis (Table H in S3 Annex) to understand associations between the steps in the sequence. For all cadres there are low-to-moderate levels of correlation (r) between providers who screen and initiate TB diagnostics (drug shops: r = 0.36; pharmacies: r = 0.36, labs: r = 0.33, clinical: r = 0.55), but there is no correlation between screening and refraining from inappropriate or unnecessary drug dispensing. A moderately-positive correlation is observed for drugs shops (r = 0.68) and pharmacies (r = 0.46) ordering an appropriate diagnostic and refraining from inappropriate or unnecessary drug

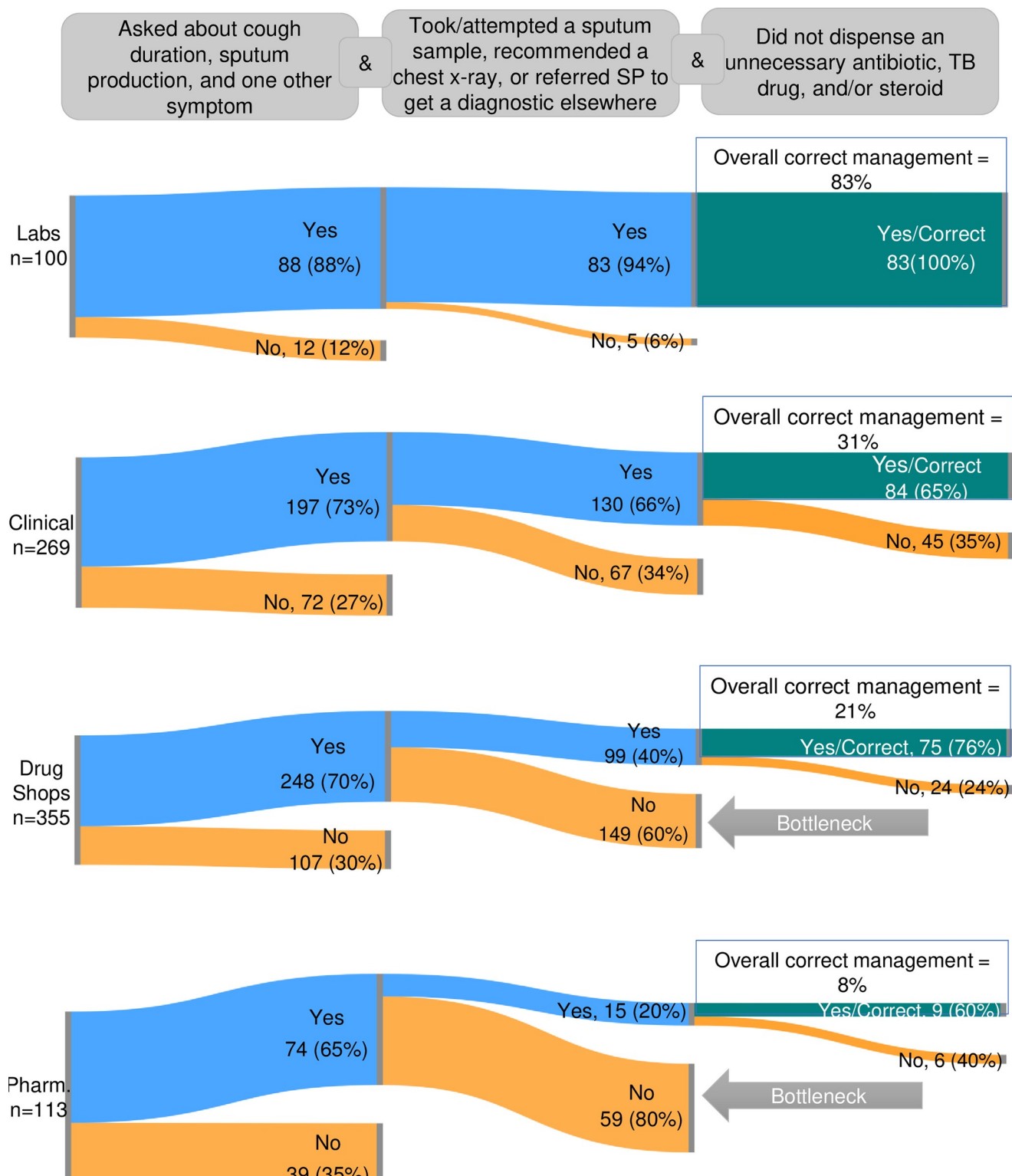

**Fig 3. Case 1: Private sector bottleneck analysis, by cadre.** The diagrams above show the expected combination of correct management actions for Case 1 for each of the private sector cadres included in this study. The orange bars depict the relative proportion of providers who exited the correct management flow at each case element. The green bars show the proportion of providers who demonstrated all three case management actions during their interaction with the SP.

**Table 3. Private provider regression results—presumptive patient scenario.**

| | Provider successfully treated SP | | | |
| --- | --- | --- | --- | --- |
| | Adjusted OR | SE | 95% CI | |
| | | | Lower | Upper |
| *Facility class/provider* | | | | |
| Clinical/Medical Doctor (reference) | | | | |
| Drug Shop/Drug Shop Proprietor | 1.33 | 0.52 | 0.62 | 2.86 |
| Drug Shop/Other Drug Shop Staff | 0.41 | 0.27 | 0.11 | 1.49 |
| Pharmacy/Pharmacist | 0.61 | 0.58 | 0.09 | 3.92 |
| Pharmacy/Other Pharmacy Staff | 0.02 | 0.03** | 0 | 0.23 |
| Laboratory/Laboratory Technician | 19.3 | 5.24*** | 11.34 | 32.85 |
| Laboratory/Other Laboratory Staff | 2.64 | 0.76*** | 1.5 | 4.65 |
| Clinical/Other Health Worker | 0.72 | 0.31 | 0.31 | 1.69 |
| *State* | | . | | |
| Kano (reference) | | . | | |
| Lagos | 1.54 | 0.36 | 0.98 | 2.43 |
| *Gender: Provider/SP* | | . | | |
| Male/Male (reference) | | . | | |
| Female/Female | 2.59 | 0.63*** | 1.6 | 4.18 |
| Male/Female | 1.76 | 0.4* | 1.13 | 2.75 |
| Female/Male | 2.42 | 0.68** | 1.39 | 4.21 |
| *Number of patients in waiting room (average at start and end of visit)* | | . | | |
| Less than 1 (reference) | | . | | |
| 1 to <2 Patients | 1.1 | 0.29 | 0.66 | 1.84 |
| 2 to <6 Patients | 0.64 | 0.18 | 0.37 | 1.1 |
| 6+ Patients | 0.86 | 0.31 | 0.43 | 1.75 |

OR: Odds ratio; SE: Robust Standard Error; CI: Confidence Interval

* $p < 0.05$

** $p < 0.01$

*** $p < 0.001$.

SP: Standardized Patient; Drug Shop: Patent and Proprietary Medicine Vendor; CP: Community Pharmacy.

dispensing, which corresponds to the bottleneck analysis finding that non-clinical providers engaging in or referring for sputum sampling tend to meet all the correct management elements.

Results from the Case 1 regression analysis (Table 3) indicate that compared to SPs who interacted with a medical doctor in a clinical facility, SPs who interacted with non-pharmacist staff in a CP were less likely to be managed correctly (adjusted OR 0.02 [0.0–0.02], p<0.01). SPs who interacted with any laboratory staff were more likely to receive correct management (laboratory technicians (adjusted OR 19.3 [11.3–32.85], p<0.001) other laboratory staff (adjusted OR 2.6 [1.5–4.6], p<0.001)) as compared to interacting with a doctor in a private clinic. Other types of providers or facilities were not statistically likely to correctly manage the SP differently than doctors. Results also show that female providers were more likely to correctly manage SPs (regardless of patient gender) than their male counterparts (female/female (adjusted OR 2.6 [1.6–4.2] p<0.001; female/male (adjusted OR 2.4 [1.4–4.2], p<0.01), and male providers were significantly more successful when interacting with female patients than male patients. Neither the state nor the number of patients waiting at the facility were strongly associated with correct SP management. Regressions for individual Case 1 elements, (Table I

in S3 Annex) reflect similar patterns to the overall correct case management regression, how-ever in the regression examining influences on whether or not a TB diagnostic was recom-mended or initiated for the SP, shop assistants in drug shops or pharmacies (rather than the pharmacist or drug shop proprietor) were significantly less likey to have initated or referred for an approporate TB diagnostic.

Finally, to contextualize the correct management rate for Case 1, we compared private SHOPS Plus providers' results to the norm for TB treatment in Nigeria, which was proxied by public clinical facilities (DOTS and non-DOTS), we include all facility cadres in this analysis since they were all trained to handle presumptive patients using the same NTBLCP curriculum and guidelines that are used in the public sector. Overall, 21% (CI: 16%-25%, SE: 2.2%) of pub-lic clinical facilities successfully treated SPs (Table J in S3 Annex). Private clinical providers (31%, CI: 28%-35%, SE:1.8%, p<0.001), laboratories (83%, CI: 80%-86%, SE: 1.5%, p<0.001), and drug shops (21%, CI: 19%-25%, SE 1.5%, p = 0.70) performed as well or better than public facilities in managing the SP in the presumptive case scenario. Pharmacies (8%, CI: 6%-10%, SE: 1.1%, p<0.001) performed worse than public facilities overall. This same pattern was observed in Lagos; Kano also had similar patterns except that the difference in performance between public (17%, CI: 12%-23%, SE: 2.7%) and private clinical facilities (22%, CI:18%-26.3%, SE: 2.1%) was not significant (p = 0.17). Tables K and L in S3 Annex for detailed Case 1 for public sector facilities.

## Case 2 results: Management of newly confirmed TB patients

Across both states, few private providers (18% [15% - 22%]) met all five of the Case 2 correct management elements (Table A in S4 Annex). Although a minority of providers met all five elements needed to demonstrate correct management of the SP, almost all (89% [86%-91%]) providers were able to demonstrate one or more of the Case 2 correct management elements. Results for individual case elements indicate that providers largely succeed in what are argu-ably the "key" steps of case management for newly diagnosed TB patients by confirming the patient's diagnosis (73% [68–77%]), refraining from dispensing non-TB drugs (83% [80–87%]), and initiating TB treatment (70% [66–74%]). A minority of providers engaged in the more "relational" elements of this case by meeting counseling-related element (e.g., explaining TB and its treatment (45% [40–50%]), and discussing the role and identification of a DOTS treatment supporter (37% [33–42%])).

The bottleneck analysis (Fig 4) reveals that among the 228 clinical providers visited by SPs for this scenario, the largest number (n = 70) of private clinical providers exited the correct management sequence by failing to provide basic counseling by sufficiently explaining TB and its treatment. Correlation analysis (Table C in S4 Annex) indicates there are moderate-to-high

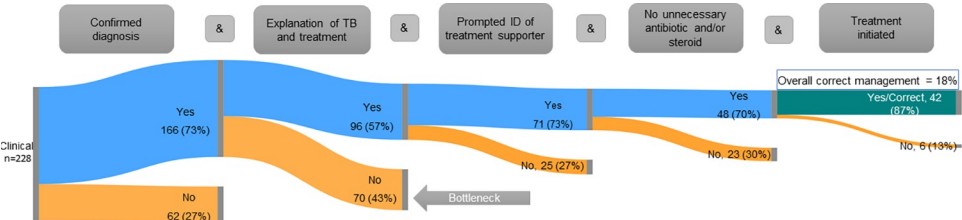

**Fig 4. Case 2: Private sector bottleneck analysis.** The diagram above shows the expected combination of correct management actions for Case 2. The orange bars depict the relative proportion of providers who exited the correct management flow at each case element. The green bars show the proportion of providers who demonstrated all five case management actions during their interaction with the SP.

levels of positive association between the different elements of the Case 2 scenario except for refraining from dispensing other non-TB antibiotics or steroids. This element had small-to-moderate negative association with the other four Case 2 elements (confirmation of diagnosis: r = -0.06; basic counseling: r = -0.37, treatment supporter discussion: r = -0.53, and treatment initiation: r = -0.23). Given the relatively small percentage of providers who dispensed inappropriate medications, this correlation is somewhat difficult to interpret in practical terms.

Unlike Case 1, there were no observable factors that were significantly associated with overall correct Case 2 management (Table D in S4 Annex). Like Case 1, female providers in Case 2 were more likely on average to demonstrate correct management, although the results were not statistically significant. Differences in performance by state were also not statistically significant for Case 2 overall, but for the individual case element of treatment initiation, significantly more private providers in Lagos initiated treatment for SPs than their counterparts in Kano. Detailed regression results for each element in the case management sequence are provided in S4 Annex File.

Finally, when compared to public DOTS facilities (Table E in S4 Annex), a proxy for the TB treatment initiation norm in Nigeria, the proportion of private clinical facilities across and within both states that correctly managed the Case 2 SP (18%, CI: 9%-20%, SE:1.9%) was not significantly different than the correct management rate observed in the public sector (13%; CI: 15%-22%, SE: 2.7%, p = 0.16).

## Discussion

This study's results show that private clinical providers, drug shops, and laboratories participating in multi-cadre PPM networks, perform at least as well as the currently available standard of care provided in public facilities. Further, when compared to results from other SP studies implemented in public and private facilities in India, Kenya, China, and South Africa [13,15,29,31], outcomes for Nigerian clinical providers compare favorably. Using the more lenient definition for correct management applied in these other SP studies (i.e., providing or recommending an appropriate TB diagnostic) correct case management in our study for private clinical providers is 56%, which is greater than observed correct case management rates in urban India (35% in private facilities), urban Kenya (33% in private facilities), rural China (35% in public facilities), and urban South Africa (43% in private facilities) (See S5 Annex).

Nevertheless, overall rates for correct case management in Lagos and Kano indicate a need to improve adherence to standard TB management protocols. Considering that the COVID-19 Global Pandemic is reported to be diverting and straining TB control programs in LMICs and is subsequently anticipated to result in increases in undetected TB cases [39–41], it is more important than ever to ensure that health care workers are efficiently and correctly identifying and managing presumptive and confirmed TB patients. This is particularly true for male providers. As our regression analysis shows, male providers were less successful in correctly managing patients in comparison to their female counterparts, especially when the SP was male. The gender disparities identified in this study dovetails with analysis of TB notification and modeling data from many other countries [42] and suggest that a considerable proportion of men never get diagnosed and started on TB treatment. This implies that TB program stakeholders would be well-served to account for gender dynamics between and among patients and providers when devising and implementing quality improvement strategies.

Although this study employs similar methods and analytical approaches as other SP studies, the addition of a bottleneck analysis allows us to pinpoint the aspects of presumptive and confirmed case management that were more difficult for providers to follow, and thus what TB

control programs, implementing partners, and policymakers might consider prioritizing when planning training and quality improvement interventions. The remainder of this section will discuss the key implications of our findings and what these suggest for future programming and research.

Bottleneck analysis showed that though two-thirds or more of providers across cadres were able to meet minimum screening standards, there was a notable drop-off in the management sequence at the point of testing, particularly for pharmacies and drug shops. Among those who screened correctly, only 40 percent of drug shops, and 20 percent of pharmacies attempted to initiate or refer for an appropriate diagnostic. These lapses in adherence to testing protocols may have several reasons, though regression analysis indicating that SPs interacting with support staff in drug shops and pharmacies were less likely to get access to an appropriate diagnostic may in part be the result of these staff not knowing what to do. Though ongoing SHOPS Plus supportive supervision is intended in part to facilitate diffusion of case management skills and protocols to other facility staff, these results may reflect the need for further step-down training in these types of facilities. Further, the reluctance to collect sputum samples by drug shops and pharmacies is perhaps understandable given that they may consider themselves more retail- than medically oriented businesses. For example, testing adherence was much higher in laboratories (94% of those who appropriately screened), where the core business is screening and administering diagnostics. Despite these organizational differences, the unwillingness of pharmacies and drug shops to even refer patients that they have screened and found to be presumptive remains less understood and may suggest that they screen to determine what to recommend for sale rather than to identify TB cases.

Although clinical facilities managed presumptive patients more successfully than drug shops or pharmacies, the fact that a third (34%) of clinical providers who appropriately screened did not also initiate a diagnostic is not optimal for reducing the overall volume of Nigeria's missing cases. Since clinical facilities do not have the same business incentives as pharmacies or drug shops, it is possible that other market or systems issues are affecting behavior. Studies conducted in Nigeria have remarked on general challenges in public laboratory logistics systems and capacity [43,44], and during this study's implementation SHOPS Plus observed (and received provider complaints about) contemporaneous issues with the public-sector-anchored testing system, including delayed sample pickups, GeneXpert cartridge stock-outs, and inadequate power supply to process samples. Considering that clinical providers who, along with their patients, likely find lengthy and unpredictable test turnaround times unacceptable, the observed bottleneck in testing adherence could reflect provider reluctance to engage with a troubled testing system that could subsequently reflect poorly on their own practices.

Outside of screening and diagnostics, results across cadres show that while inappropriate dispensing of drugs typically used to treat DS-TB occurred in relatively few presumptive cases (11% across all cadres who provide drugs), the dispensing of other broad- and narrow-spectrum antibiotics in the absence of a laboratory-confirmed infection was more common (49% across pharmacies, drug shops, and clinical facilities). These findings are in line with the well-documented concern that profit motives tend to incentivize irrational prescribing and dispensing of other types of antibiotics among private providers, especially pharmacists and drug shops [6,45,46]. However, our bottleneck and correlation analysis demonstrate that drug shops and pharmacies who adhere to the testing protocol also tend to refrain from dispensing unnecessary medications. This suggests that most providers who are actively and appropriately contributing to presumptive identification are not undermining their efforts with irrational dispensing, implying that limited resources for remediation/re-training may be best targeted to those who do not appropriately screen and initiate or refer for a TB diagnostic. Further

research on factors that drive certain private providers to be more adherent would benefit stakeholders wishing to efficiently target resources to certain cadres of private providers.

As in the presumptive TB case scenario, few clinical providers (18%) correctly managed SPs posing as confirmed TB patients in full accordance with guidelines. However, since most providers (70%) initiated treatment with SPs, they arguably succeeded in the most critical part of this case management sequence for improving Nigeria's overall treatment initiation rate. That clinical providers mainly failed to adhere to the correct protocol for counseling (i.e., describing TB treatment, its side effects, and prompting patients to identify a treatment supporter) corroborates a recent assessment of TB service quality in Nigeria, where up to 40% of patients reported they did not receive TB treatment under DOT [2]. Concerns with confidentiality and stigma, or beliefs about the reliability of a patient's social supports, could explain why providers do not encourage their patient's identification of treatment supporters [47–49]. Although failing in these areas raises concerns about facilitating patient adherence, it is also possible that providers felt that they would have opportunities to address these in future visits. Addressing these issues requires further consideration and study by program implementers and researchers.

After identifying that the main bottlenecks to correct management is adherence to testing protocols for presumptive patients and, for confirmed patients, insufficient counseling, SHOPS Plus and the Lagos and Kano state TB programs adapted the sputum transport and testing systems to be more responsive and efficient, including the use of real-time data to redirect samples away from non-functioning or backlogged laboratories. To improve counseling the program has also introduced printed and video job aides to support providers and companion printed material for patients.

## Limitations

As noted by others [50], using the SP method has its limitations. The method is limited to a one-time interaction with a provider and has not yet been validated for multiple visits to the same provider (which is typical for those on TB treatment). SPs are trained to present their symptoms and history in a manner that should allow the provider to rule out other symptomatically related conditions, but in real-life situations the clarity of patients' initial presentation varies and can negatively impact providers' clinical decision-making. SP studies avoid this situation by using standardized, pre-scripted scenarios. Finally, the SPs interacted only with staff on duty at the time of the visit, which may not have been staff who had attended formal training on managing TB patients. Since time/resource limitations reduce availability of in-service private sector training opportunities, implementers like SHOPS Plus, rely on supervision and peer relationships to diffuse protocols to other facility staff who are not otherwise able to attend formal training.

## Conclusion

SP studies are critical for understanding quality of care and identifying where improvements are required, and in Nigeria we used SPs to examine the extent to which private health providers in Lagos and Kano were in alignment with international and national standards for TB screening and treatment initiation during the implementation of the SHOPS Plus private sector engagement effort. Given that the multi-cadre feature of SHOPS Plus networks is unique in Nigeria, we also examined the variation in the quality of services provided by different facility cadres and compared them to public sector providers. Finally, this study used a novel bottleneck analysis to demonstrate the potential of the private sector's role in increasing access to high quality TB care. Although the study showed that private facilities fell short of the desired quality, the bottleneck analysis results identify specific quality issues that can be addressed by

TB program implementers with system- and provider-level interventions that are likely to be as relevant to the public sector as they are to the private sector.

The finding that private providers demonstrated a level of quality comparable to that of the public sector supports their continued inclusion in Nigeria's national TB program efforts to close the gap of unidentified TB cases and ensure timely access to diagnosis and treatment. This is of enormous importance to countries like Nigeria, where the private sector is often the first place where patients seek care and where three-quarters of TB cases go unidentified annually. The opportunity to improve TB case identification may be missed if national TB control programs and their partner organizations do not engage and strengthen these providers' capacity to participate in TB programs.

## Supporting information

**S1 Annex. Additional information on sampling, methods, and data collection.**
(DOCX)

**S2 Annex. Response rates.**
(DOCX)

**S3 Annex. Case 1 detailed results tables, including state-specific results.**
(DOCX)

**S4 Annex. Case 2 detailed results tables, including state-specific results.**
(DOCX)

**S5 Annex. Comparing results across TB quality of care studies that used SP survey designs.**
(DOCX)

## Acknowledgments

There are many who supported the successful implementation of this research, and for this we are deeply grateful. The McGill International TB Centre of McGill University, Montreal, Quebec, Canada provided SP training manuals and previously deployed data collection instruments, which we referenced and modified for this study. Ms. Mariose Amarikwa and the Qualiquant Services Inc. team were the first data collection firm in Nigeria to have successfully collected data on TB care quality using the SP method and following a complex protocol. Dr. Adebola Lawanson of the Nigeria National TB and Leprosy Control Programme, Dr. Olusola Sokoya of the Lagos State TB and Leprosy Control Programme, and Dr. Ibrahim A. Umar of the Kano State TB and Leprosy Control Promgramme provided support and approval of this study in their respective jurisdictions. Dr. Temitayo Odusote and Rupert Eneogu of the USAID Nigeria's Office of HIV/AIDS and Tuberculosis, provided programmatic support for the entire SHOPS Plus TB program. Members of the SHOPS Plus program teams in Lagos and Kano provided logistical support and feedback during SP training and data collection. Dr. Sarah Bradley of Abt Associates reviewed the protocol's data analysis plan and provided valuable comment on the draft manuscript. Maggie Stokes of Abt Associates helped prepare the article submission.

Some results from this study were previously presented at the 2020 World Union Conference on Lung Health.

## Author Contributions

**Conceptualization:** Lauren A. Rosapep, Sophie Faye, Bolanle Olusola-Faleye, Elaine M. Baruwa, Christopher Obanubi, Akinyemi Olumuyiwa Atobatele.

**Data curation:** Lauren A. Rosapep, Sophie Faye, Benjamin Johns, Ada Kwan.

**Formal analysis:** Lauren A. Rosapep, Sophie Faye, Benjamin Johns, Micah K. Sorum.

**Funding acquisition:** Bolanle Olusola-Faleye, Elaine M. Baruwa, Christopher Obanubi.

**Investigation:** Lauren A. Rosapep, Sophie Faye, Micah K. Sorum, Flora Nwagagbo, Abdu A. Adamu.

**Methodology:** Lauren A. Rosapep, Sophie Faye, Benjamin Johns, Bolanle Olusola-Faleye, Flora Nwagagbo, Ada Kwan, Akinyemi Olumuyiwa Atobatele.

**Project administration:** Lauren A. Rosapep, Sophie Faye, Bolanle Olusola-Faleye, Elaine M. Baruwa, Micah K. Sorum, Flora Nwagagbo, Abdu A. Adamu.

**Resources:** Christopher Obanubi.

**Software:** Sophie Faye, Benjamin Johns.

**Supervision:** Lauren A. Rosapep, Sophie Faye, Elaine M. Baruwa.

**Validation:** Bolanle Olusola-Faleye, Elaine M. Baruwa, Flora Nwagagbo, Abdu A. Adamu, Christopher Obanubi, Akinyemi Olumuyiwa Atobatele.

**Visualization:** Lauren A. Rosapep, Benjamin Johns.

**Writing – original draft:** Lauren A. Rosapep, Elaine M. Baruwa, Micah K. Sorum.

**Writing – review & editing:** Sophie Faye, Benjamin Johns, Bolanle Olusola-Faleye, Elaine M. Baruwa, Micah K. Sorum, Flora Nwagagbo, Abdu A. Adamu, Ada Kwan, Christopher Obanubi, Akinyemi Olumuyiwa Atobatele.

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
