## [Decision Letter · Decision Letter 0]

23 Sep 2021

 PGPH-D-21-00398 Tuberculosis care quality in urban Nigeria: a cross-sectional study of adherence to screening and treatment initiation guidelines in multi-cadre networks of private health service providers PLOS Global Public Health

Dear Dr. Rosapep,

Thank you for submitting your manuscript to PLOS Global Public Health. After careful consideration, we feel that it has merit but does not fully meet PLOS Global Public Health’s publication criteria as it currently stands. Therefore, we invite you to submit a revised version of the manuscript that addresses the points raised during the review process.

We look forward to receiving your revised manuscript.

Kind regards,

Andrew D. Kerkhoff

Academic Editor

Journal Requirements:

1. Please update the completed 'Competing Interests' statement, including any COIs declared by your co-authors. If you have no competing interests to declare, please state "The authors have declared that no competing interests exist". Otherwise please declare all competing interests beginning with the statement "I have read the journal's policy and the authors of this manuscript have the following competing interests:"

2. In the online submission form, you indicated that "All data relevant to the study are freely available to those who register as users of USAID’s Development Data Library, which can be found at: " ext-link-type="uri" xlink:type="simple">https://data.usaid.gov/Tuberculosis/Nigeria-Tuberculosis-Program-Quality-of-Care-Study/f8bn-qej9".

Reviewers' comments:

Reviewer's Responses to Questions

**Comments to the Author**

1. Does this manuscript meet PLOS Global Public Health’s publication criteria? Is the manuscript technically sound, and do the data support the conclusions? The manuscript must describe methodologically and ethically rigorous research with conclusions that are appropriately drawn based on the data presented.

Reviewer #1: Partly

Reviewer #2: Yes

2. Has the statistical analysis been performed appropriately and rigorously?

Reviewer #1: Yes

Reviewer #2: Yes

3. Have the authors made all data underlying the findings in their manuscript fully available (please refer to the Data Availability Statement at the start of the manuscript PDF file)?

Reviewer #1: Yes

Reviewer #2: Yes

4. Is the manuscript presented in an intelligible fashion and written in standard English?

Reviewer #1: Yes

Reviewer #2: Yes

5. Review Comments to the Author

Reviewer #1: This study applied standardized patient methodology to evaluate the performance of private sector providers to national guidelines for screening and trt initiation in Nigeria. Overall this is a high-quality study that was well-designed and implemented. It addressed a critical area of concern in Nigeria and globally: the quality of care received by TB patients treated in the private healthcare sector. However, I have some concerns about the utility and implementation of the bottleneck analysis in addition to several minor comments.

Major:

I'm not yet convinced of the value added by the bottleneck analysis. My concern is that the events of the ideal diagnostic cascade as outline by the authors in a monotonic sequential way (ie if you get one wrong you get all the remaining steps wrong). The bottleneck analysis disregards any provide action after their first failed step. This given the impression of very poor providers performance even though that the authors point out that for case 2 70% of providers do the most important things: initiate TB treatment. If a provider doesn't ask more questions about symptoms but does recommend a diagnostic and avoids unnecessary drugs, they're virtually invisible in the bottleneck figures.

Additionally, I am confused about why you have selected certain drop offs to be labelled as bottlenecks over others. For example in Figure 2, The 27% failure at step 1 is labelled a bottleneck but the 30% drop off at step 4 is not. What is the criteria for bottleneck?

Would a correlation analysis not provide similar data that is agnostic to the sequence of steps?

As an example of what I'm finding confusing, in line 417 the authors state "our bottleneck analysis shows that the majority of PPMVs and CPs who adhere to the testing protocol also adhere to the final stage of correct management by not dispensing unnecessary medications" But is is the rate of inappropriate dispensing difference between those who do and do not adhere to testing protocol? ie is there actually an impact of step 1 on step 3?

In short, I am not clear on what additional useful, actionable information is offered by this analysis. Currently, it muddies the water for me in light of the authors other comments that critical actions (referring for dx, or initiating trt) are relatively common.

Minor:

-Line 105, Please provide rationale for choosing Kano and Lagos in addition to the description you have provided of the setting

-line 146, the levels of the substrata are given in Table 2 but it would also be helpful to list and define them here

-line 149, what specific estimates were the sample size calculations powered for? I believe this was the MOE around the proportion of correct management?

-line 254, please provide reasons with frequency for failed interactions

-Table 3, be consistent with listing the reference category first or last (typically listed first)

-line 320, I'd prefer to see CIs for proportions over SEs

-The authors may wish as a sensitivity analysis to calculate correct management according to the lenient definition used in other SP studies to allow direct comparison to these study results

Reviewer #2: This study uses standardized patients (SPs) to evaluate TB quality of care, using methods previously developed and applied in India, Kenya, China and elsewhere. The study is well-designed, nicely reported, and well-written. The study innovates upon prior studies through its evaluation of counseling by providers and other aspects of treatment initiation in Case #2, which has previously mainly been used to evaluate diagnostic confirmation, referral, and other aspects of treatment initiation—but not counseling and other patient-centered care aspects. In addition, their analysis of gender pairings of providers and patients and implications for quality of care are very informative. The study uses a rigorous framework for stratified sampling of health facilities. I think it is a strong manuscript.

I would however, recommend the following major and minor considerations for the authors for improving the manuscript:

Major:

1. The authors should better contextualize what the findings for the private sector facilities that received the SHOPS intervention might mean for other private sector facilities in Nigeria, including the following: (a) what kind of training did the SHOPS program provide to these private sector providers on TB care, including average number of educational contact hours; (b) to what extent do the facilities enrolled in the SHOPS program represent the “average” private sector facility in these geographic areas; and (c) describe any previous studies on TB care in the Nigerian private sector or at least what is known generally about the quality of private sector care in Nigeria. This is relevant because most studies with direct comparisons of public and private sectors in LMICs have generally shown better same or better quality of TB care in the public sector. See for example, the following systematic review that evaluated quality of TB care in India and compared public and private sector provider knowledge: Satyanarayana S et al. Int J TB Lung Dis 2015;19(7):751-763, or, a more relevant comparison, a SP study of TB quality of care in Kenya, which showed similar outcomes between public/private on most metrics but better for public in some: Daniels et al. BMJ Glob Health 2017;2(2):e000333.

Therefore, the question is, how well does private sector care at SHOPS intervention sites represent private care at other sites?

2. Need to place greater weight on particular key outcomes: Overall, I appreciate the authors’ use of bottleneck analyses as shown in the figures to represent the proportion who achieve all quality of care outcomes measured. With that said, some outcomes in these analyses are more important than others. For example, in case #1, undergoing a TB-related test (sputum, Xpert, CXR) is ultimately more important for diagnosis than the other metrics (e.g., asking appropriate questions about cough, etc). Similarly, in case #2, treatment initiation is ultimately the most important outcome of that visit, compare to the other outcomes. For example, treatment initiation rate of 70% would be considered poor in many settings unless providers had an expectation that this would happen at the next visit. I think that the authors should mostly keep their current presentation of Results (the bottleneck analyses are excellent and informative), but also include a few lines that focus specifically on these key indicators and potentially conduct analyses that evaluate factors associated with completion or non-completion of these two key indicators for case #1 and case #2, since these two outcomes are so important. For example, it seems that, if the outcome of treatment initiation alone is looked at, private SHOPS clinical providers may outcome public sector providers (although they do not outperform on the composite quality of care outcome).

3. For case #1, for the evaluation of whether providers provided unnecessary medications, fluoroquinolones, steroids, and TB medications (given without diagnostic workup) definitely count as unnecessary and potentially harmful for patients with presumptive TB as they might suppress symptoms and delay TB diagnosis (especially fluoroquinolones). However, whether narrow- or broad-spectrum antibiotics should count in this category of unnecessary drugs could be debated. For sputum smear-negative TB or Xpert-negative TB diagnosis, an empiric trial of (non-fluoroquinolone, non-TB) antibiotics is actually part of the diagnostic algorithm in most places, and it is a matter of clinical practice whether providers start these antibiotics while the initial TB workup is happening or not (it would not be unreasonable to do so depending on how sick you feel the patient is). I would recommend counting FQs, TB medications, and steroids as harmful and unnecessary drugs for this outcome, but potentially exclude other antibiotics.

4. Provider gender and outcomes: This was a fantastic part of the analysis, with women outperforming men in most cases. However, in the Results and Discussion, the authors did not comment on the fact that male providers seemed to also outperform in quality when their patients were women – whereas male providers performed more poorly when evaluating male patients. This is a potentially important finding. Analyses of prevalence survey data in comparison to notification data in many countries suggest that men with TB are under-diagnosed and/or under-notified and that there a considerable proportion of men never get diagnosed and started on TB treatment, see, for example: Horton KC et al. PLOS Medicine 2016; 13(9):e1002119. The authors should comment on this finding in the Results and Discussion.

Minor comments:

1. The authors should be sure to provide the full expansion of every acronym as a foot note to every figure and/or table. For example, I found myself going back to look up what CP or PPMV stood for multiple times. Moreover, these acronyms themselves are confusing and the authors might consider using the full expansion for these terms in the text rather than the acronyms, especially since CP means “community pharmacy” but this is easy to confuse with “clinical providers.” In addition, it was not clear to me what the difference is between a CP and PPMV in the real world – they both seem to be private pharmacies, so the authors should describe how these two types of facilities are different in the background and setting section.

2. In their description of the value of SP studies, instead of referring to this as a “gold standard” methodology, it might be safer to say that this is a high quality methodology for particular quality metrics. As the authors later note in the discussion, SP methods are only really good at measuring initial or single patient interactions, and cannot measure subsequent quality of care well (e.g., during the TB treatment course, later in the diagnostic workup) – other methodologies would be higher quality for later steps.

3. Page 15, line 307: I believe the adjusted OR is reported incorrectly – should be 0.02 NOT 0.2.

4. Page 16, lines 320-327: I think it is perhaps inappropriate to compare public clinical providers to other types of private sector sites—e.g., labs, pharmacies, etc. As you note, private labs have the job of trying to get people to do tests and so are not surprisingly more keen on getting patients to get tested. Instead, would restrict this public clinical providers comparison to the comparable group in the private sector—private clinical providers.

6. PLOS authors have the option to publish the peer review history of their article (what does this mean?). If published, this will include your full peer review and any attached files.

**Do you want your identity to be public for this peer review?** For information about this choice, including consent withdrawal, please see our Privacy Policy.

Reviewer #1: No

Reviewer #2: **Yes: **Ramnath Subbaraman

---

## [Decision Letter · Decision Letter 1]

7 Dec 2021

Tuberculosis care quality in urban Nigeria: a cross-sectional study of adherence to screening and treatment initiation guidelines in multi-cadre networks of private health service providers

PGPH-D-21-00398R1

Dear Dr. Rosapep,

We're pleased to inform you that your manuscript has been judged scientifically suitable for publication and will be formally accepted for publication once it meets all outstanding technical requirements.

Within one week, you'll receive an e-mail detailing the required amendments. When these have been addressed, you'll receive a formal acceptance letter and your manuscript will be scheduled for publication.

An invoice for payment will follow shortly after the formal acceptance. To ensure an efficient process, please log into Editorial Manager at https://www.editorialmanager.com/pgph/ click the 'Update My Information' link at the top of the page, and double check that your user information is up-to-date. If you have any billing related questions, please contact our Author Billing department directly at authorbilling@plos.org.

Kind regards,

Andrew D. Kerkhoff

Academic Editor

Reviewers' comments:

Reviewer's Responses to Questions

**Comments to the Author**

1. If the authors have adequately addressed your comments raised in a previous round of review and you feel that this manuscript is now acceptable for publication, you may indicate that here to bypass the “Comments to the Author” section, enter your conflict of interest statement in the “Confidential to Editor” section, and submit your "Accept" recommendation.

Reviewer #1: All comments have been addressed

Reviewer #2: All comments have been addressed

2. Does this manuscript meet PLOS Global Public Health’s publication criteria? Is the manuscript technically sound, and do the data support the conclusions? The manuscript must describe methodologically and ethically rigorous research with conclusions that are appropriately drawn based on the data presented.

Reviewer #1: Yes

Reviewer #2: Yes

3. Has the statistical analysis been performed appropriately and rigorously?

Reviewer #1: Yes

Reviewer #2: Yes

4. Have the authors made all data underlying the findings in their manuscript fully available (please refer to the Data Availability Statement at the start of the manuscript PDF file)?

Reviewer #1: Yes

Reviewer #2: Yes

5. Is the manuscript presented in an intelligible fashion and written in standard English?

Reviewer #1: Yes

Reviewer #2: Yes

6. Review Comments to the Author

Reviewer #1: The authors have addressed my comments including a more detailed justification of the use of a bottleneck analysis and what the authors believe it adds to their analysis over previous methods.

Reviewer #2: The authors have done an adequate job of addressing the questions and criticisms raised.

7. PLOS authors have the option to publish the peer review history of their article (what does this mean?). If published, this will include your full peer review and any attached files.

**Do you want your identity to be public for this peer review?** For information about this choice, including consent withdrawal, please see our Privacy Policy.

Reviewer #1: No

Reviewer #2: **Yes: **Ramnath Subbaraman
